# The Highest Density of Phosphorylated Histone H1 Appeared in Prophase and Prometaphase in Parallel with Reduced H3K9me3, and HDAC1 Depletion Increased H1.2/H1.3 and H1.4 Serine 38 Phosphorylation

**DOI:** 10.3390/life12060798

**Published:** 2022-05-27

**Authors:** Soňa Legartová, Gabriela Lochmanová, Eva Bártová

**Affiliations:** 1Institute of Biophysics of the Czech Academy of Sciences, Královopolská 135, 612 65 Brno, Czech Republic; legartova@ibp.cz; 2Central European Institute of Technology, Masaryk University, Kamenice 753/5, 625 00 Brno, Czech Republic; g.lochmanova@seznam.cz; 3National Centre for Biomolecular Research, Faculty of Science, Masaryk University, Kamenice 753/5, 625 00 Brno, Czech Republic

**Keywords:** histone H1, histone H3, chromatin, epigenetic, nucleolus, mass spectrometry

## Abstract

Background: Variants of linker histone H1 are tissue-specific and are responsible for chromatin compaction accompanying cell differentiation, mitotic chromosome condensation, and apoptosis. Heterochromatinization, as the main feature of these processes, is also associated with pronounced trimethylation of histones H3 at the lysine 9 position (H3K9me3). Methods: By confocal microscopy, we analyzed cell cycle-dependent levels and distribution of phosphorylated histone H1 (H1ph) and H3K9me3. By mass spectrometry, we studied post-translational modifications of linker histones. Results: Phosphorylated histone H1, similarly to H3K9me3, has a comparable level in the G1, S, and G2 phases of the cell cycle. A high density of phosphorylated H1 was inside nucleoli of mouse embryonic stem cells (ESCs). H1ph was also abundant in prophase and prometaphase, while H1ph was absent in anaphase and telophase. H3K9me3 surrounded chromosomal DNA in telophase. This histone modification was barely detectable in the early phases of mitosis. Mass spectrometry revealed several ESC-specific phosphorylation sites of H1. HDAC1 depletion did not change H1 acetylation but potentiated phosphorylation of H1.2/H1.3 and H1.4 at serine 38 positions. Conclusions: Differences in the level and distribution of H1ph and H3K9me3 were revealed during mitotic phases. ESC-specific phosphorylation sites were identified in a linker histone.

## 1. Introduction

Chromatin consists of DNA wrapped around an octamer of core histone proteins H2A, H2B, H3, and H4. Genomic regions between nucleosomes are protected by linker histone H1, responsible for chromatin condensation and access of other regulatory factors to linker DNA [1]. It is well known that histones, H1, are small proteins consisting of a ∼75 residue globular domain that is flanked by the N-terminal tail (∼20–35 aa) and the C-terminal region (∼100 aa) [2]. In human cells, we know eleven H1 variants with seven somatic subtypes (H1. 1 to H1. 5, H1.0, and H1X), three testis-specific variants (H1t, H1T2, and HILS1), and one oocyte-specific H1 variant, called H1oo [3,4]. Among others, Millán-Ariño et al. showed that the H1.2 variant is less abundant in transcription start sites of transcriptionally inactive loci and is enriched in guanine-cytosine (GC)-poor genomic regions located in close proximity to lamina-associated domains [5]. Conversely, H1.0 and H1X variants are abundant in CpG islands and gene-rich genomic regions [5]. Importantly, H1.0 accumulates in terminally differentiated cells [4]. Li et al. showed the different distribution of H1.5 in human embryonic stem cells (ESCs) compared with differentiated fibroblasts [6]. Importantly, H1.5 was found in transcriptionally silent loci encoding membrane-associated proteins in differentiated cells, but it was not the case with pluripotent human ESCs.

For regulation of chromatin structure, DNA replication, and transcription, post-translational modifications (PTMs) of core histones are functionally significant, and a linker histone H1 is not an exception. The following H1 modifications were described: methylation, acetylation, ubiquitination, formylation, poly-ADP ribosylation, and phosphorylation [7]. H1 phosphorylation (H1ph) is responsible for chromatin condensation and cell cycle regulation [8]. In mammalian cells, the lowest phosphorylation level of H1 is in the G1 phase of the cell cycle, but H1ph increases in S/G2 phases, with the maximum peak in metaphase [7,9]. Interestingly, dephosphorylation of H1 variants precedes DNA oligonucleosomal fragmentation, representing a late-stage hallmark of apoptosis [10]. So, it is evident that H1 phosphorylation status is crucial in many cellular processes, including cell cycle regulation, cell differentiation, and apoptosis. Importantly, H1.2 phosphorylation also affects the p53-mediated DNA damage response [11,12], which should also be connected with the p53-dependent apoptotic pathway.

From the view of PTMs of histone H1, it is essential to mention that specific methylation sites of H1 were also detected. Ohe et al. identified Ezh2-dependent methylation of H1.4K26 in mammalian cells [13]. Due to the catalytic activity of Ezh2 (a component of the Polycomb group PRC2 complex), it is expected that this H1 modification has a silencing effect [14]. In addition to this observation, G9a histone methyltransferase (HMT), generally mediating dimethylation of histone H3 at the lysine 9 (H3K9me2), can also methylate human histone H1.4 at the position of lysine K26 in vivo. This process is essential for chromatin condensation, namely, heterochromatin formation. In addition, Terme et al. showed a functional link between H1.4K26 methylation and H1S27 phosphorylation and extended this to H1.4K26 acetylation [15]. Among others, H1.4K34 acetylation regulates the recruitment of H1 to chromatin and contributes to efficient transcription processes [16].

Based on the observation mentioned above, we tested a hypothesis of whether there is a specific nuclear arrangement of phosphorylated histone H1 in distinct cell cycle phases. Additionally, we analyzed a functional link between seemingly unrelated factors such as phosphorylated histone H1 and a marker of heterochromatin, a core histone H3 trimethylated at lysine 9. We studied if depletion of histone deacetylase 1 (HDAC1) affects the nuclear arrangement of linker histone H1 and changes the epigenetic profile of H1 variants, including H1.0, H1.1, H1.2, H1.3, H1.4, and H1.5.

## 2. Materials and Methods

### 2.1. Cultivation of Mouse Embryonic Stem Cells and Tumor Cells

We studied wild-type mouse embryonic stem cells (mESCs), D3 line (mESCs wt, wild-type) [17] and mESCs that were deficient in HDAC 1 (HDAC1 dn mESCs) [18,19]. Cells were cultivated on 0.2% gelatin-coated Petri dishes (valid for wt-cells) or Matrigel-coated plastic dishes (#354277; Thermo Fischer Scientific, Rockford, IL, USA; valid for HDAC1 dn-cells). The mouse ESCs were grown in Dulbecco’s Modified Eagle Medium (DMEM; #6429, Merck, Prague, Czech Republic) supplemented with 15% fetal bovine serum (FBS; #FB-1280, Biosera, Nuaille, France), 0.1 mM non-essential amino acids (NEAA; #11140050, ThermoFisher Scientific, Prague, Czech Republic), 100 μM 1-thioglycerol (MTG; #M6145, Merck, Prague, Czech Republic), 1 ng/mL leukemia inhibitor factor (LIF; #LIF2010, Merck, Prague, Czech Republic), 10,000 IU/mL penicillin, and 10,000 μg/mL streptomycin (#XC-A4122, Biosera, Nuaille, France).

The HeLa (human cervix adenocarcinoma) cell line (ATCC^®^ CCL-2TM, ATCC, Manassas, VA, USA) and HeLa-Fucci (expressing RFP-Cdt1 in the G1 phase and GFP-geminin in the S/G2/M phases, also see [20]) were cultivated in DMEM supplemented with 10% FBS and the appropriate antibiotics, at 37 °C in a humidified atmosphere containing 5% CO_2_.

### 2.2. Immunodetection and Confocal Microscopy of Histone Markers

Immunofluorescent analysis was performed according to [21]. Firstly, the cells were fixed in 4% formaldehyde (#AAJ19943K2, Fisher Scientific, Pittsburgh, PA, USA) for 10 min at room temperature. The next step was permeabilization with 0.2% Triton X-100 (#194854, MP Biomedicals, Santa Ana, CA, USA) for 8 min and 0.1% saponin (#S7900, Merck, Prague, Czech Republic) for 13 min. After that procedure, cells were washed twice in 1× PBS for 15 min. As a blocking solution, we used 10% goat serum (#G9023, Merck, Prague, Czech Republic) dissolved in 1× PBS-Triton X-100 (0.1%) for one hour at room temperature, followed by a washing step in 1× PBS for 15 min. The following primary antibodies were used for immunofluorescence analysis: anti-histone H1 (phospho) (#ab4270, Abcam, Cambridge, UK) and anti-H3K9me3 (#NBP1-30141, Novus Biological, Abingdon, UK). The dilution concentration was 1:100 in blocking buffer, and incubation was at 4 °C overnight.

The next day, samples were incubated with the secondary antibodies, including goat anti-mouse Cy3 (#ab97035, Abcam, Cambridge, UK) and goat anti-rabbit Cy3 (#A10520, Thermo Fisher Scientific, Waltham, MA, USA), goat anti-rabbit Cy5 (#ab6564, Abcam, Cambridge, UK), anti-mouse AlexaFluor^TM^ 647 (#32728, Thermo Fisher Scientific, Waltham, MA, USA). The cell nuclei (condensed chromatin) were visualized using 4′,6-diamidino-2-phenylindole (DAPI; #D9542, Merck, Czech Republic). The DAPI was dissolved in antifade mounting medium Vectashield (#H-1000, Vector Laboratories, Burlingame, CA, USA). DAPI enabled visualization of AT-rich regions in DNA.

We acquired images with Leica TCS SP8X SMD confocal microscope (Leica Microsystem, Wetzlar, Germany), equipped with HC PL APO 63×/1.4 oil CS2 objective. Image acquisition was performed using a white light laser (WLL) and 405 nm laser with the following excitation wavelengths (λ_ex._) for fluorophores mentioned above: Cy3 (λ_ex._ = 554 nm); Cy5 (λ_ex._ = 649 nm) and DAPI counterstain (λ_ex._ = 358 nm). DAPI was visualized by a 405 nm laser connected to Leica TCS SP8X SMD confocal microscope. The axial and lateral resolution values of HC PL APO 63×/ 1.4 oil CS2 objective are provided in Table 1 (below).

### 2.3. Proteomic Analysis of Histone H1 Phosphorylation Status

#### 2.3.1. Histone Preparation for Liquid Chromatography-Tandem Mass Spectrometry LC–MS/MS Analysis

Histones were isolated as described by [22]. Briefly, cells were washed twice with ice-cold phosphate-buffered saline (PBS), re-suspended in lysis buffer (80 mmol/L NaCl, 20 mmol/L EDTA, 1% Triton X-100, 45 mmol/L sodium butyrate, and 100 mmol/L phenylmethylsulfonyl fluoride), incubated for 20 min on ice, and centrifuged at 2000× *g* for 8 min. Pellets were re-suspended in 900 μL ice-cold 0.2 mol/L H_2_SO_4_ and incubated for 2 h, shaking at 4 °C. After centrifugation at 16,000× *g*, proteins were precipitated from the supernatant with trichloroacetic acid to a final concentration of 25% and incubated for 30 min on ice. After centrifugation at 5000× *g* for 30 min at 4 °C, the pellet was washed with 50 mmol/L HCl in acetone, then with 100% acetone, and subsequently dissolved in water. The protein concentration was determined by the Bradford assay (Bio-Rad, Hercules, CA, USA). A 150 μg portion of the histone sample was diluted with acidified acetonitrile solution (80% acetonitrile, 2% formic acid). MS Phospho-mix standards (MSP1L, MSP2L, and MSP3L; Merck, Czech Republic) were added to the samples. Phosphopeptides were enriched using the Pierce Magnetic Titanium Dioxide Phosphopeptide Enrichment kit (#88811, Thermo Fisher Scientific, USA) according to manufacturer protocol. Eluates were concentrated under vacuum to 3 μL, diluted with 90 μL of 0.1% formic acid, and purified using a Hypersep SpinTip C-18 column (Thermo Fisher Scientific, USA).

#### 2.3.2. LC–MS/MS Analysis, Database Search, and Data Evaluation

Phospho-enriched histone peptides represented by 3–4 replicates of each condition were analyzed by LC–MS/MS as described by [22]. The LC–MS/MS equipment consisted of an RSLCnano system, equipped with an Acclaim Pepmap100 C18 analytical column (3 µm particles, 75 µm × 500 mm; Thermo Fisher Scientific), coupled to an Orbitrap Elite hybrid spectrometer (Thermo Fisher Scientific) equipped with a Digital PicoView 550 ion source (New Objective) using PicoTip SilicaTip emitter (FS360-20-15-N-20-C12), and Active Background Ion Reduction Device. The mobile phase consisted of 0.1% formic acid in water (A) and 0.1% formic acid in 80% acetonitrile (B), with the following proportions of B: 1% B for 16 min at 600 nL/min to concentrate peptides, then (with a switch to 300 nL/min) 1–13% B over 20 min, 13–33% B over 25 min, 33–56% B over 20 min and 56–80% B over 5 min followed by isocratic washing at 80% B for 5 min. The analytical column outlet was directly connected to the ion source of the MS. MS data were acquired using a data-dependent strategy selecting up to the top 10 precursors based on precursor abundance in a survey scan (350–2000 *m*/*z*). The resolution of the survey scan was 60,000 (400 *m*/*z*) with a target value of 1 × 10^6^, one microscan, and a maximum injection time of 1000 ms. HCD MS/MS spectra were acquired with a target value of 50,000 and resolution of 15,000 (400 *m*/*z*). The maximum injection time for MS/MS was 500 ms. Dynamic exclusion was enabled for 45 s after one MS/MS spectrum acquisition, and early expiration was disabled. The isolation window for MS/MS fragmentation was set to 2 *m*/*z*.

The RAW mass spectrometric data files were analyzed using Proteome Discoverer software (version 1.4; Thermo Fisher Scientific, USA) with the in-house Mascot search engine to compare acquired spectra with entries in the in-house histone *Mus musculus* database. Settings for all searches included trypsin enzyme specificity and up to five missed cleavages. The following variable modifications were set for searches: methyl (R, K), di-methyl (K), tri-methyl (K), acetyl (K, protein N-term), and phosphorylation (S, T, Y). Mass tolerances of peptides and MS/MS fragments for MS/MS ion searches were 7 ppm and 0.03 Da, respectively. Manual peak labeling and calculation of the peak area corresponding to each precursor ion from the extracted ion chromatograms (XICs) were done via Skyline 21.1 software (Seattle, WA, USA). A spectral library was created using the Proteome discoverer platform (version 1.4; Thermo Fisher Scientific, USA). Only peptides with statistically significant peptide scores (*p* < 0.01) were included. Rank 1 peptides with Mascot expectation value <0.01 and at least six amino acids were considered. Peptide identifications were manually verified, and quantitative data were evaluated using Skyline 21.1 software. Precursor areas of phosphorylated histone peptides were normalized to signals of MS Phosphomix standards.

The mass spectrometry proteomics data have been deposited to the ProteomeXchange Consortium via the PRIDE [23] partner repository with the dataset identifier PXD033544.

## 3. Results

### 3.1. Localization of Phosphorylated Histone H1 and H3K9me3 Is Cell Cycle Specific

By immunofluorescence, combined with confocal microscopy, we observed in HeLa Fucci cells that the distribution and density of phosphorylated histone H1, similarly to H3K9me3, were identical in the G1, S, and G2 phases of the cell cycle (Figure 1A,B). In the interphase of wild-type mouse ESCs and HDAC1 double-knockout cells, H1ph occupied nucleoli and appeared in the nucleoplasm outside H3K9me3-dense chromocenters (Figure 2(Aa–Ac)). Interestingly, DAPI-dense DNA surrounding a compartment of nucleoli did not colocalize with phosphorylated H1 that occupied a very central region of nucleoli (Figure 2(Ac)). A low level of H1ph was on the nuclear periphery (Figure 2(Ba,Bb)) that was characterized by a high density of heterochromatin marker H3K9me3 (Figure 2(Ba,Bc)). We observed that HDAC1 deficiency did not change the nuclear distribution of H1 phosphorylated form and H3K9me3 (Figure 2(Aa–Bc)).

Next, we found the most intensive clustering of H1ph in specific regions of mitotic cells (Figure 3B). Importantly, in the prophase of mitosis, H1ph was concentrated in significant clusters (Figure 3A,B). In prometaphase, H1ph was highly dense in the whole cellular content (Figure 3B, and Appendix A). In anaphase, tiny H1ph foci occupied space outside condensed DAPI-dense chromosomal DNA (Figure 3B, and Appendix A). Additionally, in telophase, we observed a barely detectable level of H1ph, but condensed chromosomes and their DNA were surrounded by H3K9me3 (Figure 3B, and Appendix A).

### 3.2. Post-Translational Modifications of Histone H1 Studied by Mass Spectrometry

We studied post-translational modifications of histone H1 in mESC wt and HDAC1 dn mouse ES cells by mass spectrometry. We identified the following H1 isoforms: H1.0, H1.1, H1.2, H1.3, H1.4, and H1.5. In the co-translation N-terminal domain, we observed the most abundant PTMs in all H1 variants on serine S2. In detail, H1.2 was acetylated at serine 2 (S2), while H1.1, H1.3, H1.4, and H1.5 variants were found to carry combinatorial marks of N-acetylation and O-phosphorylation at S2. In the case of the H1.0 variant, we found acetylation at threonine 2 (Figure 4). Additionally, we have observed the following phosphorylated sites: H1.1 S199ph; and H1.2 S38ph, T156ph; and H1.3 T18ph, S38ph, T156ph; and H1.4 T18ph, S38ph, S199ph; and H1.5 S18ph (Figure 4). Interestingly, the acetylation profile of H1 was not affected by HDAC1 depletion, only an increase in the level of H1.2/H1.3 phosphorylation (KASphGPPVSELITK motif) and H1.4 phosphorylation (KTSphGPPVSELITK motif) we observed in HDAC1 depleted cells (Figure 5, see also Appendix A). 

## 4. Discussion

Histone H1.0 variant, specific for terminally differentiated cells, is enriched in the repetitive sequences of ribosomal genes (summarized by [24]). Moreover, Okuwaki et al. showed the enrichment of the H1.0 variant in the intergenic regions of ribosomal genes, while H1.0 is less abundant in their promoters and coding regions [25]. This observation fits well with our data showing high relative phosphorylation of histone H1 in a compartment of nucleoli studied in mESCs (Figure 2(Ab,Ac)). In general, a high density of phosphorylated histone H1 was found in genomic regions with a pronounced RNA synthesis [26]. In this case, nucleoli, as sites of ribosomal genes’ transcription, can be considered the biggest transcription factories [27]. Therefore, an occurrence of phosphorylated H1 inside nucleoli fits well with the theory of Zheng et al., showing that site-specific H1 phosphorylation in interphase facilitates transcription that is mediated by both RNA polymerases I and II. These authors observed that especially H1.2/H1.4 (e.g., pS187-H1.4) variants occupy nucleoli [26].

In prophases and prometaphase of mESCs, we have found a high level of phosphorylated H1, while anaphase and telophase were characterized by an absence of H1 phosphorylation. In the G1, S, and G2 phases of the cell cycle, the distribution profile of H1ph was identical (Figure 1 and Figure 3A,B and Appendix A). Additionally, Sarg et al. summarized that the individual H1 subtypes differ in their degree of phosphorylation during the cell cycle [28]. Zheng et al. in human cells showed specific sites in histones H1.2 and H1.4 that were phosphorylated only during mitosis or during both mitosis and interphase [26]. Green A et al. documented that interphase H1 phosphorylation is the most pronounced in the G1 or early S phase of the cell cycle [29]. Talasz et al. showed a low level of phosphorylated H1 in the G1 phase and a peak of H1 phosphorylation in the G2 phase and mitosis [9]. These authors additionally published that phosphorylation of histone H1.5 Ser17 appears early in the G1 phase, while the Ser172 phosphorylation is activated later in the G1 phase nuclei [30]. Importantly, histone H1.5 Thr10 phosphorylation exclusively occurred in mitotic cells. These data document significant distinctions in the abundance of histone H1 variants and their post-translational modifications in cell cycle phases, including mitosis.

Using mass spectrometry, in mouse ECSs, we found specific phosphorylation and acetylation sites of histone H1. In this case, we analyzed histones H1.0, H1.1, H1.2, H1.3, H1.4, and H1.5, and we observed that mainly the N-terminal domain of H1 variants is post-translationally modified (Figure 4). Additionally, Starkova et al. showed by the use of a highly precise cyclotron resonance mass spectroscopy (MALDI–FT–ICR–MS), novel post-translational modifications of H1, including meK34-mH1.4, meK35-cH1.1, meK35-mH1.1, meK75-hH1.2, meK75-hH1.3, acK26-hH1.4, acK26-hH1.3 and acK17-hH1.1 [31]. Starkova et al. published that the differences in PTMs in the N- and C-terminal tails of H1.2, H1.3, and H1.4 affected the interaction of H1 with DNA and related proteins [31].

Here, we observe in mESCs that the histone H1.0 variant is acetylated at threonine 2 on the N-terminal domain (Figure 4), but HDAC1 depletion did not change this acetylation profile. Interestingly, deficiency in HDAC1 affected phosphorylation of H1.2, H1.3, and H1.4 (Figure 5). These data support the existence of cross-talk between acetylation and phosphorylation. For example, in bacteria, van Noort et al. showed that deletion of the two N-acetyltransferases affects protein phosphorylation [32]. Additionally, Uhart and Bustos documented a link between phosphorylation and lysine acetylation in human cells. In this case, it seems to be likely that functional cross-talk between phosphorylation and acetylation could apply not only to non-histone but also to histone proteins [33].

Several studies showed that H1 phosphorylation is a feature of decondensed chromatin rather than highly condensed heterochromatin [6,34]. This claim also supports our observation that phosphorylated histones H1 appear in transcriptionally active nucleoli and in the nuclear interior that is considered to contain more relaxed and transcriptionally active chromatin [35]. We found that the nuclear periphery, abundant in silencing epigenetic marker H3K9me3, is characterized by a low density of phosphorylated histone H1 (Figure 2(Ba–Bc), arrow in the direction of the nuclear periphery). Similarly, highly condensed chromocenters (clusters of centromeric heterochromatin) were absent of H1 phosphorylation (Figure 2(Ba–Bc), arrow in the direction to selected chromocenters). These data fit well with the theory that histone H1 phosphorylation affects chromatin condensation and appears in the transcriptional active genomic region to potentiate transcription efficacy [26].

## 5. Conclusions and Future Directions

Together, we showed that the nuclear distribution of phosphorylated histone H1 and H3K9me3 was identical in the G1, S, and G2 phases of the cell cycle. In mitosis, the highest level of H1 phosphorylation was in prophase and prometaphase with a low level of H3K9me3, while H1ph was barely detectable in anaphase and especially in telophase, characterized by an appearance of H3K9me3 around mitotic chromosomes. Significantly, the N-terminal domain of H1 was post-translationally modified, and HDAC1 depletion in mESCs affected phosphorylation of H1.2, H1.3, and H1.4 histone variants. However, HDAC1 deficiency did not change the acetylation status of linker histones.

Our data imply a future direction of research in which cross-talk between histone acetylation, phosphorylation, and methylation should be investigated. We should clarify if an absence of one epigenetic mark (e.g., specific histone acetylation) could be replaced by another but seemingly unrelated epigenetic trait (e.g., histone phosphorylation).

## Figures and Tables

**Figure 1 life-12-00798-f001:**
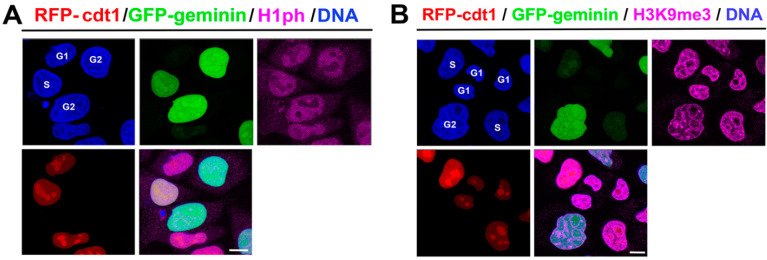
The level and distribution of phosphorylated histone H1 and H3K9me3 in interphase cell nuclei. In HeLa Fucci cells, RFP-tagged cdt1 (red) and GFP-tagged geminin (green) expression was studied. The following cell cycle phases were distinguished: G1 (RFP positivity only—red), S (a weak RFP and GFP positivity—orange), and G2 (GFP-positivity—green). An identical distribution in G1, S, and G2 cell cycle phases was observed for (**A**) phosphorylated histone H1 and (**B**) H3K9me3. (Note: H1ph was detected by the use of polyclonal antibody #ab4270, Abcam, UK) and H3K9me3 by #NBP1-30141, Novus Biological, Abingdon, UK). Scale bars show 10 µm.

**Figure 2 life-12-00798-f002:**
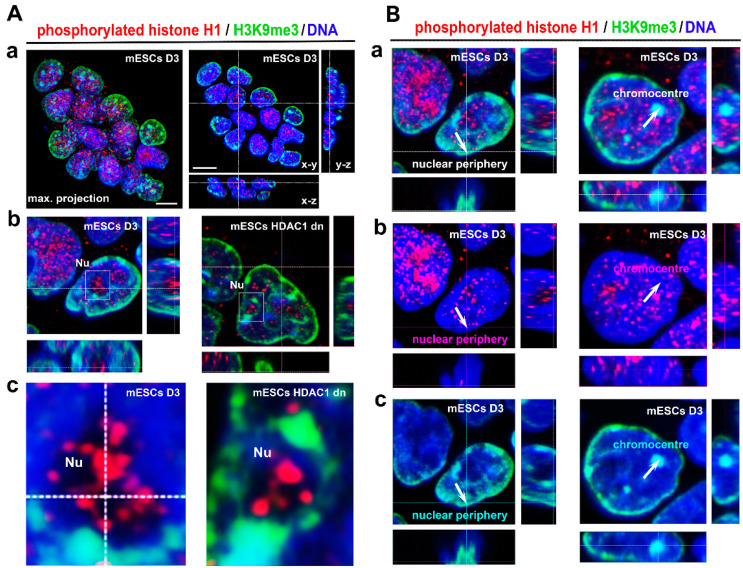
Phosphorylated histone H1 and H3K9me3 in interphase cell nuclei of HDAC1 dn mESCs and their wild-type counterpart, cell line D3. (**Aa**–**c**) Phosphorylation of histone H1 appears in the compartment of nuclei (Nu) (see magnification in panel (**Ac**)). HDAC1 depletion did not affect the nuclear distribution of H1 phosphorylated form and H3K9me3. (**Ba**) H3K9me3-dense nuclear periphery (arrow, green channel) and chromocenters (arrow, green channel) were absent of phosphorylated histone H1 (red channel) (**Bb**,**Bc**). DNA was visualized by DAPI staining and is shown in the blue channel. Scale bars in panels Aa, b, and Ba-c indicate 20 µm; in panel Ac scale bars show 1 µm.

**Figure 3 life-12-00798-f003:**
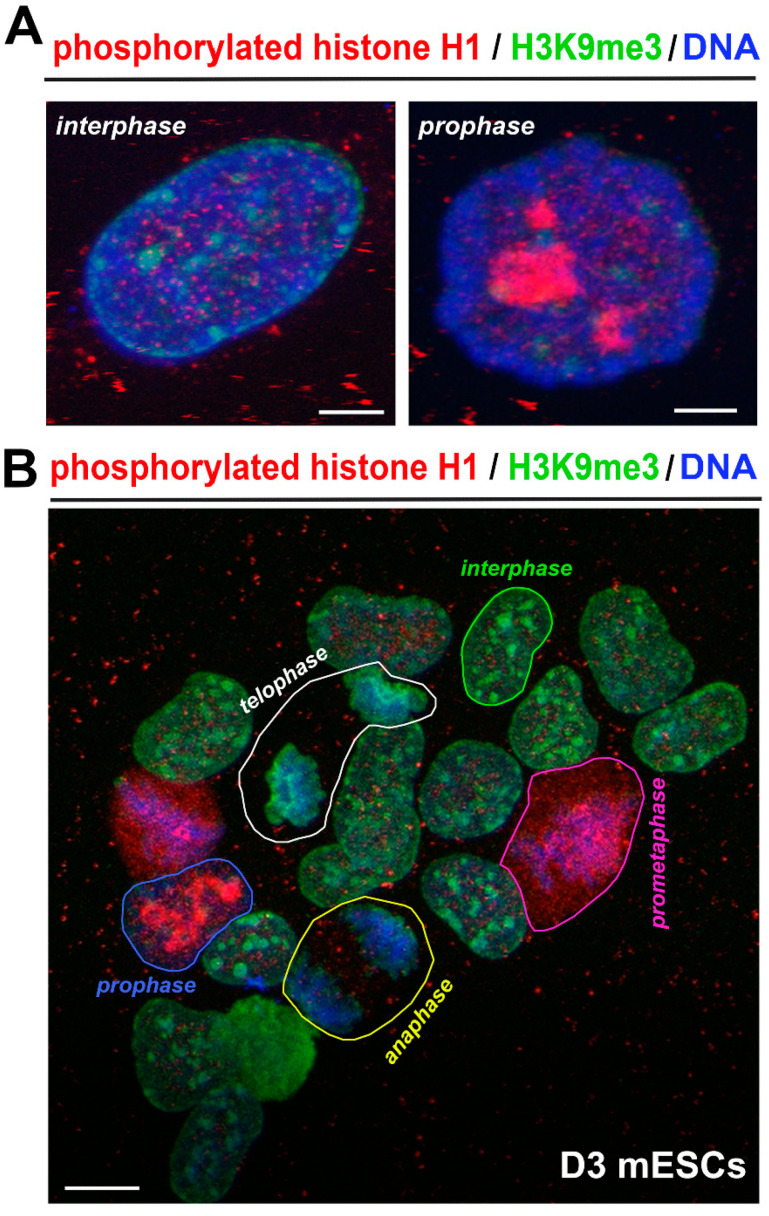
(Appendix A) Phosphorylated histone H1 (red) and H3K9me3 (green) are localized in interphase cell nuclei and mitotic cells. (**A**) Interphase is characterized by localization of H1ph in the nuclear interior. In early prophase, phosphorylated H1 is concentrated in very central nuclear regions and nucleoli. Scale bars represent 5 μm. (**B**) An example of the colony of mouse ESCs (line D3) containing cells in the following cell cycle phases: interphase (green contours), prophase (blue contour), prometaphase (magenta contour), anaphase (yellow contour), and telophase (white contour). All mitotic phases have a specific distribution pattern of phosphorylated histone H1 (red) and H3K9me3 (green). DNA was visualized by DAPI staining (blue). Scale bars represent 20 μm.

**Figure 4 life-12-00798-f004:**
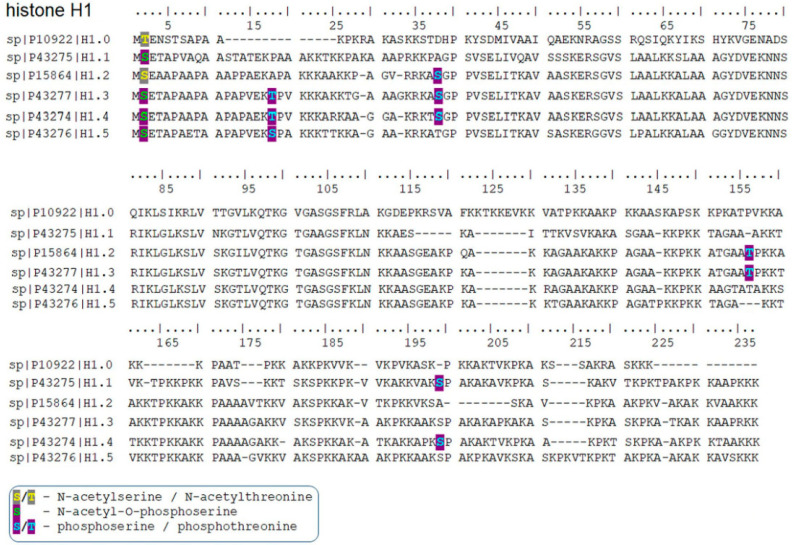
Post-translationally modified sites in histone H1 variants identified by mass spectrometry. Only highly confident peptides identified using a fixed-value PSM validator node with Mascot parameters set to Rank 1, expectation value < 0.01, and ion score ≥ 30 were considered. The following phosphorylated sites were detected by mass spectrometry: H1.1 S199ph; and H1.2 S38ph, T156ph; and H1.3 T18ph, S38ph, T156ph; and H1.4 T18ph, S38ph, S199ph; and H1.5 S18ph. N-terminal S2 and T2 of H1.2 and H1.0, respectively, were acetylated. In other H1 variants, O-phosphorylation at S2 was identified next to N-acetylation.

**Figure 5 life-12-00798-f005:**
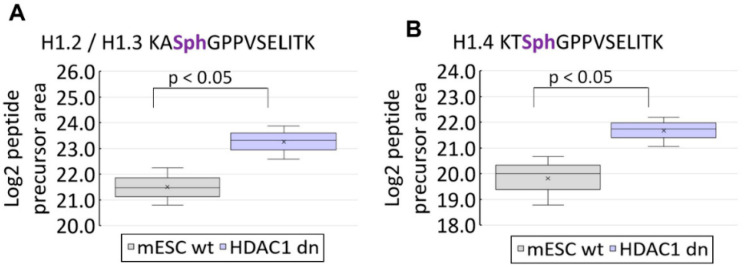
Depletion of histone deacetylases (HDAC1) in mouse ESCs increased the level of (**A**) H1.2/H1.3 phosphorylation (KASphGPPVSELITK motif) and (**B**) H1.4 phosphorylation (KTSphGPPVSELITK motif). Box-plots of the relative abundance of phosphopeptides showing extremes, interquartile ranges, means, and medians (N = 3–4). Data were normalized to the sum of phosphorylated standards. Differences between samples in normalized peptide abundances >2 at *p* < 0.05 were considered significant.

**Table 1 life-12-00798-t001:** The axial and lateral resolution values of HC PL APO 63×/1.4 oil CS2 objective for selected laser excitation wavelength (λ_ex._).

Resolution (µm)	Value (µm) for 63×/1.4 Objective for Wavelengths		
	λ_ex._ = 358 nm	λ_ex._ = 554 nm	λ_ex._ = 649 nm
Lateral resolution	0.17	0.20	0.24
Axial resolution	0.50	0.60	0.70

## Data Availability

Original micrographs (files in gigabytes, G.B.) are on demand; please address Eva Bártová (e-mail: bartova@ibp.cz or lagratova@ibp.cz). Other depository files, on-demand, see at: https://www.ibp.cz/en/about-ibp/open-data-and-it-network-rules (accessed on 10 May 2022).

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
