# Peer review of "The Highest Density of Phosphorylated Histone H1 Appeared in Prophase and Prometaphase in Parallel with Reduced H3K9me3, and HDAC1 Depletion Increased H1.2/H1.3 and H1.4 Serine 38 Phosphorylation"

_life, 2022, doi:10.3390/life12060798_

Round 1

Reviewer 1 Report

Life

Reviewers' comments:

In this manuscript, the authors evaluated the different degrees of phosphorylation in H1 during the different cell mitotic stages and the role of the histone deacetylase HDAC1 in this process. Using different biochemical methods, the authors showed that the highest levels of hyperphosphorylation are found during prophase and prometaphase showing an identical distribution in G1, S, and G2. Also, in the experiments was observed that HDAC1 influences the phosphorylation of Ser38 in H1 isoforms. This work is well written and the presentation of the results and conclusions are consistent. Therefore, I recommend this work the published in the journal Life Interactions without revisions

Author Response

Reviewers' comments:

Comments from the Editors and Reviewers:

We thank the reviewer for his/her suggestions and interesting comments regarding how to improve our manuscript. We have replied to the reviewer's criticisms. In the revised manuscript, changes are denoted in red fonts.

Reviewer I

In this manuscript, the authors evaluated the different degrees of phosphorylation in H1 during the different cell mitotic stages and the role of the histone deacetylase HDAC1 in this process. Using different biochemical methods, the authors showed that the highest levels of hyperphosphorylation are found during prophase and prometaphase showing an identical distribution in G1, S, and G2. Also, in the experiments was observed that HDAC1 influences the phosphorylation of Ser38 in H1 isoforms. This work is well written and the presentation of the results and conclusions are consistent. Therefore, I recommend this work the published in the journal Life Interactions without revisions

The answer: Many thanks, reviewer, for a positive evaluation of our results.

Reviewer 2 Report

This article is devoted to the study of the features of the intracellular distribution of phosphorylated histone H1 and trimethylated histone H3 at different stages of the cell cycle. 

Although the presented results are of scientific interest in themselves, the form of their presentation and discussion requires significant improvement. In addition, the grammar requires corrections in many places. 

Major comments.

  1. What is the reason to use the term "hyperphosphorylation" instead of traditional "phosphorylation"?
  2. It is desirable to make the abstract more clear, logical and informative without repeated statements.
  3. Lines 20-21 - repetion of the statement about absence of H1 phosphorylation in telophase.   
  4. Lines 32-38. The description of the history of the development of the histone nomenclature is not relevant to the research conducted. It is proposed to delete this paragraph.
  5. Lines 58-59. Wrong statement and reference to [14] "deposphorylation ... induced apoptosis... " According to [14] "deposphorylation was observed... 45 min after apoptosis induction". 
  6. Lines 61-62. Wrong statement "H1.2 phosphorylation triggers 
    the p53-mediated DNA damage response [15]". I did not find any phrase about the relationship between p53, H1.2 and its phosphorylation in the cited review. 
  7. Please check all your statements for compliance with what is really written in the cited articles.
  8.    Subsection 2.4. No description of instrumentation used. Description of mass spectrometry experiments is insufficient. 
  9. Experimental data on mass spectrometry are not presented in any form in the article. How can readers judge the reliability and significance of the results discussed and the differences identified? 
  10. Line 257, Figure 4, lines 286-287. - a set of contradictions regarding H1.0.  
  11. There is an evident contradiction between FRET data (Fig. 6) and statements like "On the other hand, we showed that the H3K9me3-dense nuclear periphery was absent of hyperphosphorylated histone H1.... Similarly, highly condensed chromocenters (clusters of centromeric heterochromatin) were absent of H1 hyperphosphorylation..." Distribution of FRET signal shows opposite result.
  12.  Line 325-326. "phosphorylation on specific sites of H1 is distinct..." - Experimental data do not show this.  
  13. The sections "discussion" and "conclusion" are written very haphazardly and do not give a clear idea of the key results obtained in the article and the biological significance of the detected changes.
  14. Check the grammar: lines 15-17, lines 18-20,  lines 80-82, lines 264-265, etc. Check the entire text carefully.

Minor comments

1. Subsections 2.2 and 2.3. The exact excitation wavelengths should be indicated, not a general specification of the white light laser. 

2. Line 124 " 400 Hz, bidirectional mode, and zoom 4-8" bring no useful information. Instead,  lateral and axial resolution should be indicated. 

The same about resolution at lines 147-148. 

3. Line 135. TO-PRO-3 Iodide is not an antibody.  

4. Figure 4 shows aa seqencies, not "mass spectrometry"

5. Lines 300-302. Whose data is being discussed? Authors? Literary data?

6. Table 1.   R0×QYCy5 data should be deleted. *1e15  - this value should be typed  correctly. 

Author Response

Reviewers' comments:

Comments from the Editors and Reviewers:

We thank the reviewer for his/her suggestions and interesting comments regarding how to improve our manuscript. We have replied to the reviewer's criticisms. In the revised manuscript, changes are denoted in red fonts.

Reviewer II

This article is devoted to the study of the features of the intracellular distribution of phosphorylated histone H1 and trimethylated histone H3 at different stages of the cell cycle. 

Although the presented results are of scientific interest in themselves, the form of their presentation and discussion requires significant improvement. In addition, the grammar requires corrections in many places. 

Major comments.

  1. What is the reason to use the term "hyperphosphorylation" instead of traditional "phosphorylation"?

The answer: This term was corrected to phosphorylation

  1. It is desirable to make the abstract more clear, logical, and informative without repeated statements.

The answer: An abstract was improved.

  1. Lines 20-21 - repetition of the statement about absence of H1 phosphorylation in telophase.   

The answer: This part of the abstract was corrected.

  1. Lines 32-38. The description of the history of the development of the histone nomenclature is not relevant to the research conducted. It is proposed to delete this paragraph.

The answer: We deleted this paragraph.

  1. Lines 58-59. Wrong statement and reference to [14] "deposphorylation ... induced apoptosis... " According to [14] "deposphorylation was observed... 45 min after apoptosis induction". 

The answer: We corrected this reference.

  1. Lines 61-62. Wrong statement "H1.2 phosphorylation triggers 
    the p53-mediated DNA damage response [15]". I did not find any phrase about the relationship between p53, H1.2 and its phosphorylation in the cited review. 

The answer: This was a wrong citation selected automatically by EndNote. The proper citation is Kim K, Jeong KW, Kim H, Choi J, Lu W, Stallcup MR, An W. Functional interplay between p53 acetylation and H1.2 phosphorylation in p53-regulated transcription. Oncogene. 2012 Sep 27;31(39):4290-301. doi: 10.1038/onc.2011.605. Epub 2012 Jan 16.

Please check all your statements for compliance with what is really written in the cited articles.

The answer: In the revised version, statements were verified in the literature sources

  1.    Subsection 2.4. No description of instrumentation used. Description of mass spectrometry experiments is insufficient. 

The answer: We briefly described our protocol on mass spec in the revised chapter 2.4. A detailed description of sample preparation, LC-MS/MS and data evaluation has been added to Supplementary Data.

  1. Experimental data on mass spectrometry are not presented in any form in the article. How can readers judge the reliability and significance of the results discussed and the differences identified? 

The answer: In the revised version of the manuscript, raw mass spectrometry data have been uploaded into the Pride database (Username: [email protected] <mailto:[email protected]>, Password: YzSGvC3i).

Line 257, Figure 4, lines 286-287. - a set of contradictions regarding H1.0.  

The answer: This part was corrected, and the description of PTMs of H1 variants was corrected. We apologize for such mistakes.

There is an evident contradiction between FRET data (Fig. 6) and statements like "On the other hand, we showed that the H3K9me3-dense nuclear periphery was absent of hyperphosphorylated histone H1.... Similarly, highly condensed chromocenters (clusters of centromeric heterochromatin) were absent of H1 hyperphosphorylation..." The distribution of the FRET signal shows the opposite result.

The answer: This FLIM-FRET results should be contradictory just only for the first view. Of course, it is evident that a high density of H3K9me3 is at heterochromatin appearing at the nuclear periphery and at chromocenters, but in the rest of the genome, we have also silenced genes whose promoters can be H3K9 trimethylated. Moreover, Keenan et al. doi: https://doi.org/10.1101/2020.08.13.249078 showed that the nuclear lamina-tethering of Suv39-dependent H3K9me3 domains provides an essential scaffold to support euchromatic genome organization and the maintenance of gene transcription for healthy cellular function. So this phenomenon can be a reason why we have a relatively high FLIM-FRET positivity for H1ph and H3K9me3. This observation was also very surprising for us. We expected much higher FLIM-FRET efficiency for H1ph and DNA in comparison to H1ph and H3K9me3. Also, Liao et a doi: 10.1186/s13072-017-0135-3 showed that H1 phosphorylation is dynamic in a site-specific and gene-specific manner, especially in pluripotent cells induced to differentiation. Importantly, an enrichment of pS187-H1.4 at specific genes contributes to their transcription. This part was mentioned in the revised version.

Line 325-326. "phosphorylation on specific sites of H1 is distinct..." - Experimental data do not show this.  

The answer: This sentence was corrected. We observed that H1ph is distinct in cell cycle phases, especially in mitosis phases.

  1. The sections "discussion" and "conclusion" are written very haphazardly and do not give a clear idea of the key results obtained in the article and the biological significance of the detected changes.

The answer: The discussion and conclusion sections were improved.

Check the grammar: lines 15-17, lines 18-20,  lines 80-82, lines 264-265, etc. Check the entire text carefully.

The answer: Grammar was verified by Grammarly software and by a native speaker.

Minor comments

  1. Subsections 2.2 and 2.3. The exact excitation wavelengths should be indicated, not a general specification of the white light laser. 

This part was corrected in the revised version.

  1. Line 124 " 400 Hz, bidirectional mode, and zoom 4-8" bring no useful information. Instead,  lateral and axial resolution should be indicated. 

This part was corrected in the revised version.

  1. Line 135. TO-PRO-3 Iodide is not an antibody.  

Answer:  The reviewer is absolutely right. We used TO-PRO-3 Iodide stain to study the interaction between H1ph (donor) and DNA (acceptor) for its similar far-red fluorescence to Cy5.  This part was corrected in the revised version of the manuscript.

  1. Figure 4 shows aa sequences, not "mass spectrometry"

The answer: This part was corrected; many thanks for the remark. The legend in Fig. 4 was rewritten.

  1. Lines 300-302. Whose data is being discussed? Authors? Literary data?

The answer: Line 300-302 is empty; it is a figure position, but we revised all statements around these lines.

  1. Table 1.   R0×QYCy5 data should be deleted. *1e15 this value should be typed correctly. 

The answer: This part was corrected in the revised version.

Round 2

Reviewer 2 Report

In the revised version authors eliminated a number of shortcomings, but the manuscript still is not ready for publication. It still requires major revision

Major comments.

Abstract

  1. It is quite desirable to revise the abstract and make it clear and logical.

In the rules to Life journal it is written:

Abstract: The abstract should be a total of about 200 words maximum. The abstract should be a single paragraph and should follow the style of structured abstracts, but without headings: 1) Background: Place the question addressed in a broad context and highlight the purpose of the study; 2) Methods: Describe briefly the main methods or treatments applied. Include any relevant preregistration numbers, and species and strains of any animals used. 3) Results: Summarize the article's main findings; and 4) Conclusion: Indicate the main conclusions or interpretations. The abstract should be an objective representation of the article: it must not contain results which are not presented and substantiated in the main text and should not exaggerate the main conclusions.

 Please, follow this instruction.

  Authors like to use introductory words at the beginning of sentences: similarly, also, additionally, “for example, till now”, etc. But these introductory words disturb logic and the meaning of the text in many cases. Please, check and correct it.

Lines16-17: “chromosomes in telophase are wrapped by H3K9me3” – please, think about what you have written and how you can prove and explain it.

Abbreviations like H3K9me3 should be decrypted in the Abstract.

“In interphase, a degree of interaction we found between H1ph and core 17 H3K9 trimethylation (me3) to a similar extent as shown by FLIM-FRET for the phosphorylated histone H1 and DNA”. – I do not understand this sentence.

“an intervention to the acetylome affected the phosphorylation status of histone H1” – this statement is too general and not clear.

“Phosphorylated histone H1 similarly to H3K9me3 has an identical density in G1, S, and G2 phases of the cell cycle” – this result is not mentioned in the Abstract. Why?

Results

  1. Lines 225-226. “Moreover, a low level 225 of H1ph was on the nuclear periphery”.  This statement cannot be verified with Figure 2 because three images (green, red and blue) were merged. Please show each distribution in a separate panel (for example, as a supplementary figure).
  2. In Figures 2 and 3, three colors were used: green, red and blue. Origin of a blue color is not described. I suppose, it is a distribution of DNA that is shown in blue. If so, I do not understand, why authors do not describe features of mutual distribution of DNA (chromatin) and H1ph, DNA and H3K9me.  
  1. Lines 312-313. “The presence of described modifications of H1 variants was not affected by HDAC1 depletion.”  - It is desirable to clarify this statement. What about acetylation?  
  2. Several questions arise concerning presentation and interpretation of FRET-FLIM data.

“we detected (16.6. ± 2.1) % of FRET efficiency” – please, clarify how this value was obtained. Is it FRET efficiency averaged over the nucleus area or over the area with non-zero FRET?

Figure 6. Distributions of Cy3-H1ph (A) and Cy5-H1ph (B) looks different: granular in the first case and rather diffuse in the last case. Why?

Since two images (green and red) were merged in Fig.6, it is not clear how many “green” signal is under “red” signal and vice versa in the regions, where FRET was detected.  It would be more informative to show also individual distributions of H1ph, DNA and H3K9me3 in separate panels. 

Two scales are shown near FRET panels (Events and Efficiency). What exactly is shown in the FRET panels?  A map of FRET efficiency values in different areas of a nucleus? A number of events in each point of the nucleus with the indicated FRET value (i.e. E[%]=16.6+-2.1)? As I understand it is impossible to show both Events and Efficiency in the same image.         

Please, make some biological and/or functional conclusions from the data obtained in FRET-FLIM experiments or delete them. Currently, just a Figure is shown without any interpretations.

Discussion

 According to the Life-journal rules authors should discuss the results and how they can be interpreted in perspective of previous studies and of the working hypotheses. The findings and their implications should be discussed in the broadest context possible and limitations of the work highlighted. Future research directions may also be mentioned. 

From my point of view, in the discussion section, the authors simply listed the results obtained during their study and listed previously published results without discussion and biological interpretation.

Conclusions

Please try to formulate the conclusions more clearly and convincingly.

Lines 397-400. Is there really a contradiction between data on the identical distribution  of H1ph and  H3K9me3 in the G1, S, and G2 phases of the cell cycle   and data of Talasz et al.? Authors did not examined relative phosphorylation of H1 subtypes in the G1, S, and G2 phases. Moreover, distribution of H1ph is not identical, it is qualitatively similar   in the G1, S, and G2 phases of the cell cycle.    

Minor comments

  1. Do not introduce abbreviations if you do not use them at least three times in the text. For example, LADs, HMT, H3K9me2, etc.
  2. Line 50 “dephosphorylation of H1 variants also appeared”- why "also"?
  3. Line 116, Table 1 (λ = 461 nm)– error. λex = 405 nm?
  4. Lateral resolution in the recorded confocal images depends on the numerical aperture of the objective, excitation wavelength and a size of a pixel in the digitized image. Last parameter  was not taken into account, when authors  provided information about resolution.
  5. Many of the designations in the footnote to table 2 are not described (FD, QD, ε); “Föster” -  misprint. 
  6. Subsection 2.4.1. contains many misprints in units: mmol.L 1, 900 L, H2SO4, etc.
  7. FA, MS, LC-MS/MS – abbreviations should be initially decrypted. 
  8. Lines 224 – change “HDAC1 double knockouts” to, for example, “cells with HDAC1 double knockouts”  
  9. Lines 274-275. “Chromocenters (arrow) were absent of phosphorylated histone H1; a low density of H1ph was found in these nuclear regions” – contradiction.
  10. Line 296 “Phosphorylated histone H1 (red) and H3K9me3 (green) are localised in interphase and mitosis.” – check the meaning.
  11. Line 320-321 “…were recognized…. were observed.” – check the grammar.
  12. Lines 387-390 - check the grammar.
  13. Lines 350-353: also… also….also.
  14. Figure 4. The differences in the color scheme used to designate N-acetylserine and N-acetyl-O-phosphoserine are insignificant. Please make them more distinguishable.

Author Response

We thank the reviewer for his/her suggestions and interesting comments regarding how to improve our manuscript. We have replied to the reviewer's criticisms. In the revised manuscript, changes are denoted in red fonts.

In the revised version authors eliminated a number of shortcomings, but the manuscript still is not ready for publication. It still requires major revision

Major comments.

Abstract

  1. It is quite desirable to revise the abstract and make it clear and logical.

In the rules to Life journal it is written:

Abstract: The Abstract should be a total of about 200 words maximum. The Abstract should be a single paragraph and should follow the style of structured abstracts, but without headings: 1) Background: Place the question addressed in a broad context and highlight the purpose of the study; 2) Methods: Describe briefly the main methods or treatments applied. Include any relevant preregistration numbers, and species and strains of any animals used. 3) Results: Summarize the article's main findings; and 4) Conclusion: Indicate the main conclusions or interpretations. The Abstract should be an objective representation of the article: it must not contain results which are not presented and substantiated in the main text and should not exaggerate the main conclusions.

The answer:  Revised abstract was structured into Background, Methods, Results, and Conclusion. In total, it is 200 words. We used such a heading because we saw the same in some papers published in the Life journal. If it is not essential to use these headings, they can be easily deleted.

We did not use animal models; thus, no additional information is provided in the revised version.

 Please, follow this instruction.

 Authors like to use introductory words at the beginning of sentences: similarly, also, additionally, "for example, till now", etc. But these introductory words disturb logic and the meaning of the text in many cases. Please, check and correct it.

The answer: In the original version, we used 3x “for example”. Such introductory formulation we deleted in the revised version. Also, till now was deleted. We revised the whole text and tried to eliminate such introductory words.

Lines16-17: "chromosomes in telophase are wrapped by H3K9me3" – please, think about what you have written and how you can prove and explain it.

Abbreviations like H3K9me3 should be decrypted in the Abstract.

The answer: The reviewer is right that this sentence (lines 16-17) is wrong. We changed it to DNA of chromosomes in telophase was surrounded by trimethylated histones H3 at K9. The abbreviation H3K9 was explained in the Abstract.

"In interphase, a degree of interaction we found between H1ph and core 17 H3K9 trimethylation (me3) to a similar extent as shown by FLIM-FRET for the phosphorylated histone H1 and DNA". – I do not understand this sentence.

The answer: Based on the reviewer's suggestion and an additional assessment of FLIM data, these data were deleted in the revised version.

"an intervention to the acetylome affected the phosphorylation status of histone H1" – this statement is too general and not clear.

The answer: This sentence was rewritten.

"Phosphorylated histone H1 similarly to H3K9me3 has an identical density in G1, S, and G2 phases of the cell cycle" – this result is not mentioned in the Abstract. Why?

The answer: We did not realize that this sentence was not mentioned in the Abstract. So, in the revised version, this part was mentioned.

Results

  1. Lines 225-226. "Moreover, a low level 225 of H1ph was on the nuclear periphery".  This statement cannot be verified with Figure 2 because three images (green, red and blue) were merged. Please show each distribution in a separate panel (for example, as a supplementary figure).

The answer:  We make this figure clearer. In the revised version, we separated channels and showed RGB, RB, and GB channels. In this revised version, the nuclear periphery and chromocenters are better visible, similarly to an absence of H1ph (red) on the nuclear periphery (blue contours of the cell nuclei) and H3K9me3-dense (green) nuclear periphery (see revised Fig. 2Bb, Bc).

  1. In Figures 2 and 3, three colors were used: green, red and blue. Origin of a blue color is not described. I suppose, it is a distribution of DNA that is shown in blue. If so, I do not understand, why authors do not describe features of mutual distribution of DNA (chromatin) and H1ph, DNA and H3K9me.  

The answer: The reviewer is correct that DNA is stained by DAPI (visualized in blue spectra); and thus, labeled in blue color. Figs 2 and 3 were revised. Also, a link between DNA and H1ph or DNA and H3K9me3 was mentioned in the revised Results section.

  1. Lines 312-313. "The presence of described modifications of H1 variants was not affected by HDAC1 depletion."  - It is desirable to clarify this statement. What about acetylation?  

The answer: The acetylation profile shown in Fig. 4 was not surprisingly affected in HDAC1 depleted cells. HDAC1 depletion and its effect on H3K9ac we published recently in Vecera et al., JPC, 2018.

  1. Several questions arise concerning presentation and interpretation of FRET-FLIM data.

The answer: In the revised version, we deleted FLIM/FRET data

  1. "we detected (16.6. ± 2.1) % of FRET efficiency" – please, clarify how this value was obtained. Is it FRET efficiency averaged over the nucleus area or over the area with non-zero FRET?

The answer: In the revised version, we deleted FLIM/FRET data

  1. Figure 6. Distributions of Cy3-H1ph (A) and Cy5-H1ph (B) looks different: granular in the first case and rather diffuse in the last case. Why?

The answer: It is a question of fluorochrome used. Based on this criticism, we decided to delete FLIM/FRET data because the results did not bring any additional valuable information on H1 functioning.

Since two images (green and red) were merged in Fig.6, it is not clear how many "green" signal is under "red" signal and vice versa in the regions, where FRET was detected.  It would be more informative to show also individual distributions of H1ph, DNA and H3K9me3 in

Two scales are shown near FRET panels (Events and Efficiency). What exactly is shown in the FRET panels?  A map of FRET efficiency values in different areas of a nucleus? A number of events in each point of the nucleus with the indicated FRET value (i.e. E[%]=16.6+-2.1)? As I understand it is impossible to show both Events and Efficiency in the same image.      

Please, make some biological and/or functional conclusions from the data obtained in FRET-FLIM experiments or delete them. Currently, just a Figure is shown without any interpretations.

The answer: We deleted FLIM/FRET data in the revised version, as the reviewer suggested.

Discussion

 According to the Life-journal rules authors should discuss the results and how they can be interpreted in perspective of previous studies and of the working hypotheses. The findings and their implications should be discussed in the broadest context possible and limitations of the work highlighted. Future research directions may also be mentioned. 

The answer: We improved the discussion section in the revised version and figured out future directions.

From my point of view, in the discussion section, the authors simply listed the results obtained during their study and listed previously published results without discussion and biological interpretation.

The answer: The discussion section was improved.

Please try to formulate the conclusions more clearly and convincingly.

The answer: We tried to improve the Conclusions in the revised version.

Lines 397-400. Is there really a contradiction between data on the identical distribution  of H1ph and  H3K9me3 in the G1, S, and G2 phases of the cell cycle   and data of Talasz et al.? Authors did not examined relative phosphorylation of H1 subtypes in the G1, S, and G2 phases. Moreover, distribution of H1ph is not identical, it is qualitatively similar   in the G1, S, and G2 phases of the cell cycle.   

The answer: The reviewer is entirely correct that Talasz et al. studied epitope-specific phosphorylation of H1.5 at G1-phase, S-phase, and mitosis of the cell cycle. Thus, the distribution of relative H1ph studied in our case is not comparable with the observation of these authors. It is a pity that the producer of antibody against H1ph (#ab4270, Abcam, Cambridge, UK) did not specify H1 phosphorylation sites. Together, the antibody against H1ph used in our experiments revealed a qualitatively similar level of H1ph in the G1, S, and G2 phases of the cell cycle (Fig. 1A).

 Minor comments

  1. Do not introduce abbreviations if you do not use them at least three times in the text. For example, LADs, HMT, H3K9me2, etc.

The answer: OK

  1. Line 50 "dephosphorylation of H1 variants also appeared"- why "also"?

The answer: "Also" was deleted

  1. Line 116, Table 1 (λ = 461 nm)– error. λex = 405 nm?

The answer: Corrected to 350 nm (the excitation maxima of DAPI, with the use of 405 nm laser line)

  1. Lateral resolution in the recorded confocal images depends on the numerical aperture of the objective, excitation wavelength and a size of a pixel in the digitized image. Last parameter  was not taken into account, when authors  provided information about resolution.

The answer: We corrected  Table 1. The zooms and scanning steps were set so that each pixel covers an x-y area ~ 65 nm × 65 nm

  1. Many of the designations in the footnote to table 2 are not described (FD, QD, ε); "Föster" -  misprint. 

The answer: FLIM- FRET data were deleted in the revised version.

  1. Subsection 2.4.1. contains many misprints in units: mmol.L 1, 900 L, H2SO4, etc.

The answer: Methodological part 2.4. was inspected.

  1. FA, MS, LC-MS/MS – abbreviations should be initially decrypted. 

The answer: We revised the abbreviation, and for example, the method, LC-MS/MS was clarified  - liquid chromatography-tandem mass spectrometry.

  1. Lines 224 – change "HDAC1 double knockouts" to, for example, "cells with HDAC1 double knockouts."

The answer: Corrected  

  1. Lines 274-275. "Chromocenters (arrow) were absent of phosphorylated histone H1; a low density of H1ph was found in these nuclear regions" – contradiction.

The answer: This part was corrected.

  1. Line 296 "Phosphorylated histone H1 (red) and H3K9me3 (green) are localized in interphase and mitosis." – check the meaning.

The answer: We revised this part.

  1. Line 320-321 "…were recognized…. were observed." – check the grammar.

The answer: We revised words like recognized and observed.

  1. Lines 387-390 - check the grammar.
  2. Lines 350-353: also… also….also.

The answer: The discussion section was revised by a native speaker.

  1. Figure 4. The differences in the color scheme used to designate N-acetylserine and N-acetyl-O-phosphoserine are insignificant. Please make them more distinguishable.

The answer: The labeling of Fig. 4 was improved – see revised version.

Round 3

Reviewer 2 Report

Manuscript can be recommended for publication after minor revision.

Abstract.

1. It is proposed to change line 17 as follows.

..., we analyzed cell cycle-dependent level and distribution of  phosphorylated histone H1 (H1ph) and H3K9me3.

2. It is proposed to change lines 23-25 as follows.

Mass spectrometry revealed several ESC-specific phosphorylation sites of H1. HDAC1 depletion did not change H1 acetylation but potentiated phosphorylation of H1.2/H1.3 and H1.4 at serine 38 positions.

3. It is proposed to change lines 26-27 as follows.

Differences in the level and distribution of H1ph and H3K9me3 were revealed during mitotic phases.  ESC-specific phosphorylation sites were identified in a linker histone. 

 Introduction

4. Line 43. Delete  “In this case”

5. Line 67-68. “This process is essential for chromatin condensation respectively, heterochromatin formation.”

It is not clear. Change “respectively” to “, namely,”?  

 Materials and Methods

6. Lines 118-127.

I am not sure that Leica TCS SP8X SMD confocal microscope has a laser with the emission of 350 nm, which as stated was used to excite DAPI. Please, clarify. Why do you mention “405 nm laser”?

 Discussion

 7. Line 301. “the enrichment H1.0 variant” -  may be “the enrichment of H1.0 variant” ?

8. Lines 312-313. It is not clear: “In mitotic phases of mESCs, we have found a distinct abundance of phosphorylated  H1, while is interphase, the distribution profile of H1ph was identical”       

“In mitotic phases of mESCs, we have found a distinct abundance of phosphorylated  H1” -   is this statement true for telophase and anaphase?

“while is interphase” – change “is” to “in”.  

“In interphase, the distribution profile of H1ph was identical” – identical to what?

Author Response

We thank the reviewer for his/her valuable comments. After a careful inspection of our manuscript, we revised all criticized points properly and made changes in the revised text. Changes are shown in red fonts.

Comments and Suggestions for Authors

Manuscript can be recommended for publication after minor revision.

Abstract.

  1. It is proposed to change line 17 as follows.

..., we analyzed cell cycle-dependent level and distribution of  phosphorylated histone H1 (H1ph) and H3K9me3.

The answer: OK

  1. It is proposed to change lines 23-25 as follows.

Mass spectrometry revealed several ESC-specific phosphorylation sites of H1. HDAC1 depletion did not change H1 acetylation but potentiated phosphorylation of H1.2/H1.3 and H1.4 at serine 38 positions.

The answer: OK

  1. It is proposed to change lines 26-27 as follows.

Differences in the level and distribution of H1ph and H3K9me3 were revealed during mitotic phases.  ESC-specific phosphorylation sites were identified in a linker histone.

The answer: OK

 Introduction

  1. Line 43. Delete  “In this case”

The answer: OK

  1. Line 67-68. “This process is essential for chromatin condensation respectively, heterochromatin formation.”

It is not clear. Change “respectively” to “, namely,”?  

The answer: OK

 Materials and Methods

  1. Lines 118-127.

I am not sure that Leica TCS SP8X SMD confocal microscope has a laser with the emission of 350 nm, which as stated was used to excite DAPI. Please, clarify. Why do you mention “405 nm laser”?

The answer: We apologize for incorrect information. DAPI λex/λem = 358/461 nm https://www.thermofisher.com/order/catalog/product/D1306), and we visualize DAPI by the use of a 405-nm laser connected to the Leica SP8 confocal microscope. 

 Discussion

  1. Line 301. “the enrichment H1.0 variant” -  may be “the enrichment of H1.0 variant” ?

The answer: We apologize for this mistake. Even the Grammarly advanced linguistic software did not catch this mistake.

  1. Lines 312-313. It is not clear: “In mitotic phases of mESCs, we have found a distinct abundance of phosphorylated  H1, while is interphase, the distribution profile of H1ph was identical”       

“In mitotic phases of mESCs, we have found a distinct abundance of phosphorylated  H1” -   is this statement true for telophase and anaphase?

“while is interphase” – change “is” to “in”.  

“In interphase, the distribution profile of H1ph was identical” – identical to what?

The answer: This part was corrected in the revised version.

The revised version of the manuscript was corrected by Grammarly software.
